# A Deep Learning Method for the Detection and Compensation of Outlier Events in Stock Data

**Vashalen Naidoo** [†] and **Shengzhi Du** *,[†]

Department of Electrical Engineering, Tshwane University of Technology, Pretoria West, Pretoria 0183, South Africa
* Correspondence: dushengzhi@gmail.com
† These authors contributed equally to this work.

**Abstract:** The stock price is a culmination of numerous factors that are not necessarily quantifiable and significantly affected by anomalies. The forecasting accuracy of stock prices is negatively affected by these anomalies. However, very few methods are available for detecting, modelling, and compensating for anomalies in financial time series given the critical roles of better management of funds and accurate forecasting of anomalies. Time series data are a data type that changes over a defined time interval. Each value in the data set has some relation to the previous values in the series. This attribute of time series data allows us to predict the values that will follow in the series. Typical prediction models are limited to following the patterns in the data set without being able to compensate for anomalous periods. This research will attempt to find a machine learning method to detect outliers and then compensate for these detections in the prediction made. This concept was previously unimplemented, and therefore, it will make use of theoretical work on market forecasting, outliers and their effects, and machine learning methods. The ideas implemented in the paper are based upon the efficient market hypothesis (EMH), in which the stock price reflects knowledge about the market. The EMH hypothesis cannot account for consumer sentiment towards a stock. This sentiment could produce anomalies in stock data that have a significant influence on the movement of the stock market. Therefore, the detection and compensation of outliers may improve the predictions made on stock movements. This paper proposes a deep learning method that consists of two sequential stages. The first stage is an outlier detection model based on a long short-term memory (LSTM) network auto-encoder that can determine if an outlier event has occurred and then create an associated value of this occurrence for the next stage. The second stage of the proposed method uses a higher-order neural network (HONN) model to make a prediction based on the output of the first stage and the stock time series data. Real stock data and standalone prediction models are used to validate this method. This method is superior at predicting stock time series data by compensating for outlier events. The improvement is quantifiable if the data set contains an adequate amount of anomalous periods. We may further apply the proposed method of compensating for outliers in combination with other financial time series prediction methods to offer further improvements and stability.

**Keywords:** LSTM—long short-term memory; auto-encoder; outlier detection; outlier compensation; stock price prediction

## 1. Introduction

The primary function of a stock market is to provide for the capital needs of a business. The valuation of a business or its stock price is based on availability and demand, which are heavily influenced by a multitude of factors of the business, such as intellectual property, fixed assets, liquidity, and cash flow [1]. These factors contribute to the complexity of predicting the movement of the stock market. The correct and timely prediction of the stock market could allow for the possibility of forecasting changes in an economy and

supplement competitive advantages to stakeholders [2]. Predictive analytics requires the use of statistical techniques, such as data mining and machine learning, for the determination of trends [3]. We could use the application of predictive analytics as a basis for the development of a model to find trends in the closing price of a stock market [4]. The stock market is inherently volatile, adding to the difficulty of prediction [1]. This work seeks to compensate for this volatility by adjusting the prediction accordingly. We have adapted this concept from the theoretical framework established by P. Theodossiou and A. Theodossiou [5]. By understanding the effect of outliers on the stock market, a practical method to allow a model to adapt to these effects and produce an improved prediction can be made [6]. The method is unique in its application to financial time series with only traditional predictive models to validate against.

Financial time series outlier events are events that do not follow the normal trajectory of the series. Outlier events are considered random in nature and cannot be predicted. It is then preferable to detect these events at such a time that we may act upon them [7]. An outlier event in a financial time series could be categorized by the rising or falling of the series on a steeper gradient, time-triggered events, and reversals in trends. A typical application of detecting outlier events is to find possible market manipulation and/or insider trading. Outlier events could be used to determine the true trajectory of financial markets as a function of both the regular component of the stock price summed with the product of an outlier event and its probability of occurrence [5].

Deep learning models can take large amounts of data and learn complex data trends with multiple layers of complexity. Deep learning models are further able to analyze the effects of outlier events on the resultant prediction. Stock financial time series is a complex time sequence with different scales of fluctuations, resulting in a challenge in making accurate forecasts [8]. Long short-term memory (LSTM) is capable of learning order-dependent series to produce a prediction, with it being applied in natural language processing, speech recognition, and financial time series. LSTM auto-encoders have shown resilience in the detection of anomalies in these time series thanks to a configurable threshold [9].

This paper proposes a deep learning method to compensate for the effects of outliers on stock price prediction. The data required are sourced from a public platform, Yahoo Finance, that supports the real-time interrogation of data via a software application. To validate the proposed method, the New York Stock Exchange (NYSE) is selected as a representative stock market. An LSTM auto-encoder is used to process the incoming data. This form of deep learning model finds patterns in time series data and learns from these patterns without human intervention or explicit programming (unsupervised). Two separate stages compose the finished system. In the first stage, anomalies are detected, and then a continuous output is generated. The second stage uses the output from the first stage and the closing price to make a forecast of the next closing price in the series. The second stage comprises a higher-order neural network (HONN) to find intricate patterns in the data from the LSTM auto-encoder concerning the closing price and to produce a compensated output with a low loss. Historical data on stock markets are available publicly. Outlier events that occurred in the markets are identified from published events that influence the company under study. Because a publicly traded company "Must promptly disclose to the public any material information reasonably expected to affect the value of its securities or influence an investor's decision" [10], the reported events form a reliable basis for determining an outlying period in the company's history.

After the data are processed with outlier events identified, the artificial intelligence (AI) model, LSTM, is fine-tuned. The determination of an outlier from the auto-encoder stage is accomplished by analyzing the loss function of the LSTM model by a threshold. This loss function is a continuous time series of the volatility inherent in the data. The loss function, combined with the closing price, is used as input into a fully connected higher-order neural network to predict the next day's stock price. The analysis of the results shows the LSTM auto-encoder detected anomalies (outliers) that negatively affect the AI model in predictions. Through the compensation of outliers in the prediction, the final stage

obtained a validation loss of 0.0021 for the test data. The result indicates that it is possible to compensate for outlier events to make improved predictions of time series data.

The main contributions of the paper include the modelling of outliers and compensation for their negative effects on stock price prediction. This creates a template for the detection of outliers in other time series prediction applications. The rest of the paper is organized as follows. A summary of the research used to develop this method is shown in Section 2. The method and materials that were used are discussed in Section 3, leading to the presentation of our results in Section 4. Following that, Sections 5 and 6 provide a brief discussion and conclusion regarding the results.

## 2. Related Works

### 2.1. Financial Time Series Data

Financial time series data contains complex trends that are composed of multiple polynomials with a systemically low signal-to-noise ratio that makes them difficult to meaningfully forecast [11]. There are abundant publicly available market data typically containing a minimum of six data points: the opening price, closing price, daily high, daily low, daily change, and available shares. The complex nature of stock markets means that the artificial neural network model chosen must be able to account for the volatility of the market, have no bounding in value, and be able to compensate for outlying events [12].

### 2.2. Use of Artificial Intelligence in Stock Prediction

The advent of artificial intelligence and artificial neural networks in particular have shown significant resilience in predicting financial time series events [11]. Studies into the prediction of stock market prices have yielded two main competing theories, namely the efficient market hypothesis (EMH) and the random walk approach [9]. The efficient market hypothesis states that the current market itself contains all the information about it, and if any updated information is gathered, it is absorbed by the market and reflected in its price [13], while the random walk approach says that fluctuations are random and cannot be predicted with any reasonable degree of accuracy. Proceeding on the assumption of the EMH theory being correct, making a prediction is possible, though difficult. The use of artificial intelligence and deep learning systems has simplified the modelling of nonlinear systems with a reasonable degree of accuracy [13]. Full deeplearning models are still not readily being applied as forecast models; however, simpler models based on artificial neural networks have had success in predicting short-term to medium-term fluctuations of the stock markets [14].

### 2.3. Artificial Neural Networks (ANN)

An artificial neural network is a type of system where elements of the system are derived from the functioning of the human brain, namely the neuron. These networks can modify their internal behaviour depending on the results achieved [15]. Traditional areas in which artificial neural networks (ANN) are known to excel are pattern recognition, pattern matching, and mathematical function approximation [16]. An ANN is comprised of processing elements that have inputs from other elements and/or the environment as well as its output. The connections between elements have strengths associated with each connection (how heavily one neuron is influenced by another); the strengths can change over time to better match the objective of the system [15]. This process of adjusting the strength of connections between elements is what allows the system to learn its environment.

### 2.4. Higher-Order Neural Networks (HONN)

Due to some of the limitations of traditional artificial neural networks, a higher-order network can be used to compensate for global extremes, intricate associations, and discontinuities in data. These networks have a stronger learning and storage capacity with a superior computational ability compared to the existing traditional neural networks [16]. Higher-order neural networks are a natural extension of single-layer models to more so-

phisticated ones. A complex network can be used to resolve more complex tasks such as signal processing, pattern recognition, control, and decision-making in complex systems, etc. [17].

### 2.5. Recurrent Neural Networks (RNN)

Recurrent neural networks are suited to capturing relationships in sequential or temporal data, with two derivations, in particular, showing significant promise, namely the gated recurrent neural network (Gated RNN) and long short-term memory (LSTM) recurrent neural networks [18]. RNNs' application in sequential data of varying lengths arises from their ability to have a memory function. However, this type of neural network is computationally slow and susceptible to the vanishing gradient problem in deep network architectures [19]. Gated recurrent neural networks have been applied extensively with success in the following applications: speech recognition, music synthesis, natural language processing [20]. Long short-term memory and gated recurrent unit (GRU) RNNs are very successful with long-sequence applications. What differentiates these types of recurrent neural networks is the control mechanism of updating the activation functions of elements. The control mechanism depends on both the present input to the element and the previous memory of the element. This control mechanism (or gate) is adaptively updated in the learning phase of the model and verified during the evaluation stage. Despite the advantages of being able to successfully model sequential data, these networks require increased computational power due to their gate network [21].

### 2.6. Transfer Learning

Transfer learning is a machine learning method in which one model is reused in a second model to improve the latter model. Information or parameters can be transferred between two models. This approach is commonly applied to shorten training time, handle insufficient data and improve a model's performance. A typical application of transfer learning is to shorten the training time of complex models such as those in the natural language processing space. Pre-trained models provide a starting point for the training process and reduce overall training time. This method additionally reduces the amount of data that is required to fully train a model, enabling us to model sparse datasets. This concept could be extended further to improve the accuracy of the second model when using sparse data [22].

### 2.7. Outlier Events in Stock Data

Outlier events are often overlooked due to the averaging effect of the stock market; however, a severe event could result in a catastrophic loss of wealth in a brief period [6]. A stock market trend can be broken down into its normal component, which models the averaging effect of the market. This component is subject to the efficient market hypothesis as well as an outlier component. The outlier component can be divided into multiple terms. This approach has similarities to those applied in risk management, in which each outlier event has a trigger event, a qualitative value (probability and impact of an outlier), and an event scaler [5]. The detection of outliers could be fundamentally broken down to gradient changes in the trend of a stock. The changes in gradient concerning severity, direction, and periodicity allow for the identification of specific outlier events. A possible issue with detecting outlier events is the noise in the data. Artificial neural networks have difficulty in discriminating actual events from noise and are prone to overfitting; hence, any ANN approach should be effectively tuned and of a low resolution to succeed [23].

### 2.8. Auto-Encoders

An auto-encoder is a type of neural network that is a type of unsupervised learning algorithm for dimensionality reduction. The auto-encoder uses a neural network to compress the input and then recreate it from the compressed data to make a representation of the original input. This is useful as the auto-encoder can learn the core patterns and

relationships in a dataset and then detect deviations from this pattern. An auto-encoder has two main components: an encoder component that reduces the input data dimensions to a minimum resolution without losing the core features and a decoder component that takes a compressed dataset and expands it to the dimensions of the original input data [12].

*2.9. Comparison of Similar Works*

In Table 1, we compare the merits and demerits of similar works. There are, however, numerous works on forecasting models; hence, only similar models have been analysed.

**Table 1.** Summary of related works.

| Reference | Merits | Demerits |
|---|---|---|
| [9] | A high-level analysis of deep learning approaches to stock price prediction. | An empirical study of general methods hampers the optimisation of any single approach. |
| [2] | A survey of machine learning techniques comparing the type, time frame and input variables. The study shows the advantages of machine learning in forecasting financial time series. | The survey only seeks to provide a correlation between the machine learning techniques used and their success in forecasting financial time series. |
| [21] | This paper experiments on different versions of neural networks to create a benchmark of performance. | The approaches made only allow for a comparison between the experiments done in isolation. |
| [3] | The methodology of using a simple multilayered dense model to forecast stock prices is established. | The approach made does not use state-of-the-art methods. |
| [16] | This is early research into the applications of different architectures of HONNs for financial time series forecasting. | The performance of the HONNs used is not able to match the modern deep networks. |
| [24] | This paper shows a modern method to improve the prediction accuracy of stock markets using a hybrid ARIMA and GRU network. The results are a strong indicator of the approach's success. | The method made does not handle outliers well and is constrained by the complexity of the source data. |
| [22] | The LSTM neural network approach made was found to be very accurate, with a testing root-mean-square error of 0.0859. | The model was not able to react in a timely manner to outliers events. |
| [25] | A multivariate application of machine learning using up to ten input variables to determine the direction of the movement of a stock. | The feature engineering approach made does not improve the results sufficiently to ignore the sequential nature of the data. |
| [14] | A presentation and analysis of the practicality of predicting stock prices using stock prices. | The approach made sacrifices the accuracy of the prediction for generalisation and practicality. |
| [1] | The generalised neural network architecture approach taken shows a strong result for the model's ability to assimilate different stock trends. | By creating a general model, the accuracy of the prediction was sacrificed. |
| [23] | A clustering algorithm is used to determine outliers and improve long-term prediction of stock prices. | The innovation of the work is focused on finding outliers and not improving the prediction using the events detected. |

## 3. Materials and Methods

### 3.1. Proposed Method

The method proposed to improve the accuracy of stock price prediction is based on the established theoretical framework of outliers and their effect on stock prices. The essence of this body of knowledge was used to develop our approach. It was determined that the true trajectory of the stock price could be broken into the normal trajectory and the associated probability of outliers. Thus, our method would require the detection of outliers in the stock price and then adjusting the stock price prediction based on the events detected. To detect outliers, the model would need to learn the core trend in the trajectory of the stock price. Deviations from this pattern would constitute an outlier (with some degree of error). The outlier function produced by the first model could now be used in combination with the true stock price to make an adjusted prediction of the coming stock price.

### 3.2. System Diagram of Proposed Method

The full system diagram is shown in Figure 1. The system was composed of the LSTM auto-encoder, which produces a recreated sequence used to detect outliers, and an associated loss function. A higher-order network was then able to take in the loss function and a time-associated instance of the closing stock price to produce a single prediction.

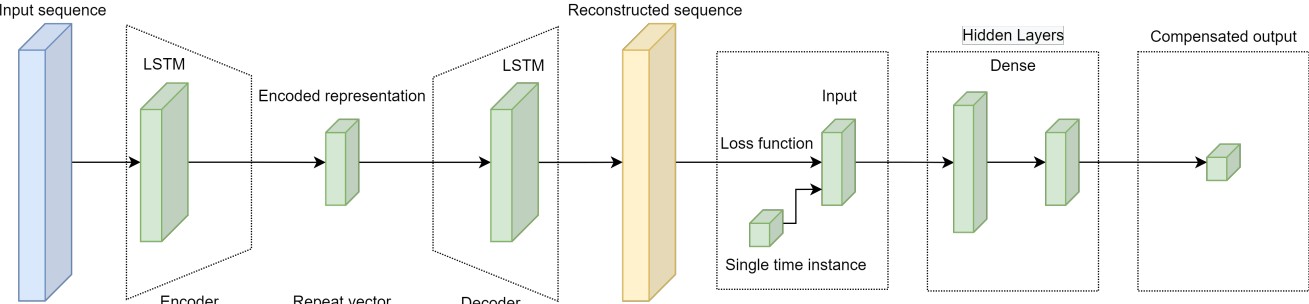

**Figure 1.** System diagram of the proposed method.

### 3.3. Transfer Diagram of Proposed Method

The information transfer that occurs between the two models is shown diagrammatically in Figure 2. The information was transferred from the outlier detection model to the prediction model as characterised by the data flow lines to create the compensated output. The stock data sets used as inputs to the two stages were sub-sets of the same data set.

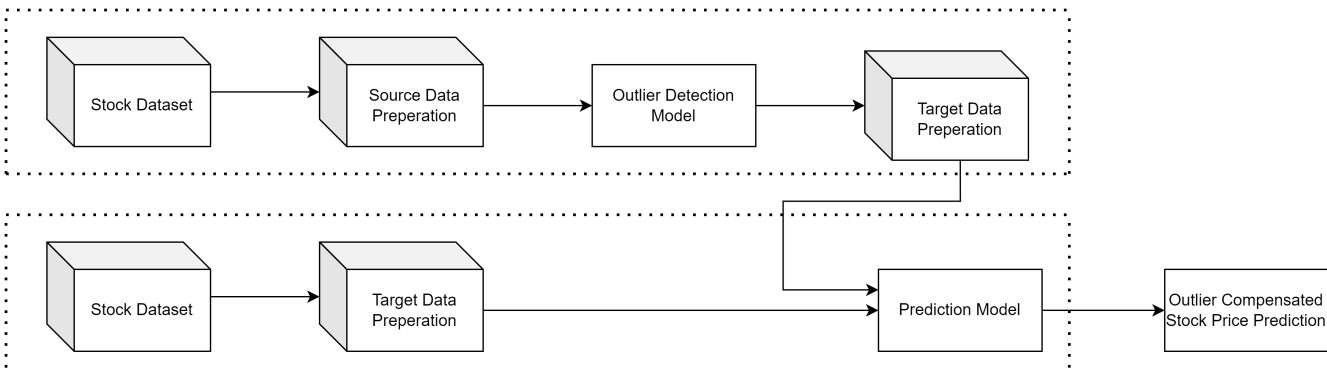

**Figure 2.** Information transfer diagram of the proposed method.

### 3.4. Data for Outlier Detection

In this paper, historical data were used to find possible trends that occur in the closing price. The technical variables available to be used in the model included the opening price, closing price, trading high, trading low, adjusted closing price and the volume of shares. Through feature analysis, technical variables were selected that showed the strongest correlation to the desired output [26]. This reduced the computational complexity of the model and improved performance [27]. The required data were sourced from Yahoo Finance, which supports the interrogation of data by a software application. Using Yahoo Finance, the NYSE was selected as the stock market for training and testing the proposed method. To create a suitable trained model, data from 1995 were considered. Using large publicly traded companies means that anomalous events are recorded publicly and readily available. Anomalous events were selected from the public archive of the companies selected for analysis [10]. Combining the data of anomalous events with the stock price provided us with labelled outlier events for fine-tuning the hyperparameters of the model.

### 3.5. Approach for Outlier Detection

Before the selection of a model for outlier detection and compensation can be made, the learning features in the available data must first be extracted. There must be a general correlation between input data features and the target variable to improve modelling performance.

Lag in a time series refers to a defined time gap of time and provides for the phenomenon of auto-correlation. Auto-correlation is the tendency of instances in a time series to correlate with the previous instance [28]. To determine the required lag of the input variable, an open-source software library called TSFresh was utilized.

The closing price was selected as the primary input variable due to its strong correlation with the forecast for the next day's closing price. This is due to the data's seasonality and data availability. The input variable for the first stage was selected to use a thirty-day lag based on the auto-correlation and seasonality of the data. The second stage input variables were determined to have a single day's lag. This showed the strongest correlation with the multivariate input data and simplified the model. The chosen lags are the basis for each stage of the model development [28].

All input variables used in the models must be scaled (a standard Keras scalar was employed in this application) [29,30]. Other variables also correlated with the target variable, but only the closing price was used in the LSTM auto-encoder. To build an LSTM auto-encoder, the input sequence (based on the input lag and variable/s) must be encoded to create a representation of the original data and then decoded to create a reconstructed sequence. The encoder must be able to reduce the input data dimensions into an encoded representation. This representation should have the smallest possible dimensions. The decoder will then take this representation and attempt to reconstruct the original sequence with the same length as the original input. This is to allow the model to learn the core patterns in the data within the lag through dimensionality reduction. The core pattern was recreated at the output of the decoder stage, which allows for any events that do not follow the core pattern to be identified [31]. This is depicted in Figure 3.

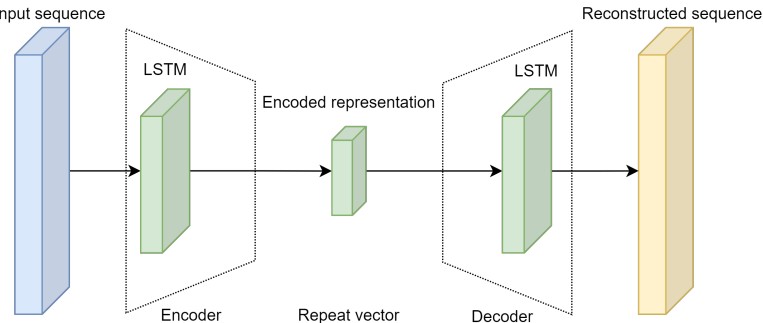

**Figure 3.** A high-level representation of an LSTM auto-encoder.

An LSTM unit cell can be visualised using the image shown in Figure 4. The unit cell shows the cell states (c), the output (h) and the input (x). The interactions between different states (previous, present and future) are also visualised.

From Figure 4, we may then determine a mathematical relationship from the output to the inputs of each LSTM cell. First, we determine the forget gate function ($f_t$) shown in Equation (1). In this equation, we have two inputs, $h_{t-1}$ and $x_t$.

$$f_t = \sigma(W_f \cdot [h_{t-1}, x_t] + b_f) \tag{1}$$

We now determine the new information to be added to the cell state (the input gate) using the function $i_t$ and $r_t$. The input variables are the same for both functions, but each

has different associated weights, biases and activation functions. The function of $i_t$ is shown in Equation (2), and $r_t$ is shown in Equation (3) [18].

$$i_t = \sigma(W_i \cdot [h_{t-1}, x_t] + b_i) \tag{2}$$

$$r_t = \tanh(W_r \cdot [h_{t-1}, x_t] + b_r) \tag{3}$$

We are now able to determine the cell states from Equations (1)–(3). This is shown in Equation (4).

$$c_t = f_t c_{t-1} + i_t r_t \tag{4}$$

We are now able to determine the result of the output gate $o_t$ using the inputs $h_{t-1}$ and $x_t$. This is shown in Equation (5).

$$o_t = \sigma(W_0[h_{t-1}, x_t] + b_0) \tag{5}$$

This provides us with the means to determine the output of the cell $h_t$ using the result of the output gate $o_t$ and the cell's current state $c_t$, as shown in Equation (6).

$$h_t = o_t \tanh C_t \tag{6}$$

The variant of LSTM here is a standardised form of the family, with other variants differing slightly from the approach made here.

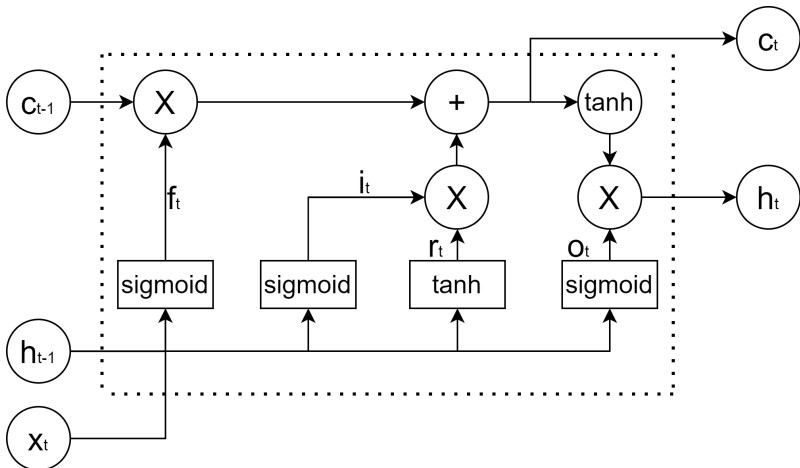

**Figure 4.** An LSTM unit cell.

The model was composed of a total of two LSTM layers with 128 neurons each; two dropout layers with the probability of a neuron dropping out set to 20 percent; a single repeat vector to compress the input sequence to 30 samples; and a single TimeDistributed-Dense layer, which applies the same Dense operation to every time-step of a 3D tensor to produce a single output. These are arranged sequentially as in Figure 3, with the encoder and decoder sections comprising an LSTM and dropout layer each. The corresponding algorithm for this stage is shown in Algorithm 1 with a corresponding model summary.

---

**Algorithm 1** An LSTM auto-encoder

---

**Require:** The closing stock price as of 1985

　1: Initialization of required libraries;

　2: Load stock data;
　3: Removal of invalid data:
　4: Scale data using the Keras Standard Scalar;
　5: Split data into training and test sets;

　6: Create an LSTM auto-encoder model;
　7: Use the training data to train the LSTM auto-encoder;
　8: Evaluate the model;

　9: **while** Evaluation result improves **do**
10: 　　Adjust the hyperparameters of the model;
11: **end while**

12: Evaluate the distribution of the training loss function;
13: Create a threshold from the training loss distribution;
14: Evaluate the threshold against the test loss distribution;
15: Collaborate the test results with labelled outlier events;

16: **while** all outlier events are not detected by the model **do**
17: 　　Adjust the threshold of the model;
18: **end while**

19: Produce a final detection of outliers from the model as a function;

---

The algorithm was implemented on top of the TensorFlow library using Keras. The model was executed using Google Colab with hardware acceleration using a Tesla K80 graphics processing unit (GPU). The processing time required (*s*) to process the data was 1.476 *s*; the time required to train the model was 1006.233 *s*, and the time required to execute the model was 7.552 *s*. Our implementation method was not intended to perform a computational complexity analysis of the algorithm but to allow for the evaluation of the concept. The model code for the LSTM auto-encoder is shown in Listing 1.

Listing 1: LSTM auto-encoder model

```python
model = Sequential()
#Encoder section
model.add(LSTM(128, activation='relu',
          input_shape=(X_train.shape[1], X_train.shape[2])))
model.add(Dropout(rate=0.2))

#Encoded representation
model.add(RepeatVector(X_train.shape[1]))

#Decoder section
model.add(LSTM(128, return_sequences=True))
model.add(Dropout(rate=0.2))

#Output layer
model.add(TimeDistributed(Dense(X_train.shape[2])))

#Optimizer and loss function
```

```
optim = tf.keras.optimizers.SGD(learning_rate=25e-4)
model.compile(optimizer=optim, loss='mse')

model.summary()
```

The resultant parameters of the LSTM auto-encoder are shown in Table 2.

**Table 2.** The LSTM auto-encoder model summary.

| Layer (Type) | Output Shape | Parameters |
|---|---|---|
| LSTM 1 | (None, 128) | 66,568 |
| Dropout 1 | (None, 128) | 0 |
| Repeat Vector | (None, 30, 128) | 0 |
| LSTM 2 | (None, 30, 128) | 131,584 |
| Dropout 2 | (None, 30, 128) | 0 |
| Time Distributed Dense | (None, 30, 1) | 129 |

The model parameters shown in Table 2 show the type of layer, the shape of the layer and the number of editable parameters in each layer. The parameters were adjusted during the training phase to match the data features.

### 3.6. Anomalies

Periods of uncertainty have been tabulated in Table 3 based on published annual reports from General Electric and articles on General Electric's business. These events were used to fine-tune the outlier detection algorithm.

**Table 3.** Publicly available events that may influence the value of General Electric stock.

| Date | Events |
|---|---|
| 2008 January | Federal Open Market Committee lowered rates in response to falling home sales rate. |
| 2008 March | The Fed began bailouts. |
| 2008 April | The Fed lowered the rate. |
| 2008 May | The Fed auctioned another USD 150 billion through the Term Auction Facility. |
| 2008 July | Wall Street's fears make it more difficult for private companies to raise capital. |
| 2008 September | Fannie and Freddie bailout. |
| 2008 September | Lehman Brothers bankruptcy. |
| 2008 October | Congress passed the USD 700 billion bank bailout bill. Central Banks Coordinate Global Action. |
| 2008 November | The Fed restructured its aid package. GM, Ford and Chrysler Request Bailouts. |
| 2008 December | The Treasury inserted USD 105 billion in TARP funds into eight banks in return for preferred stock. |
| 2009 February | The company slashed its yearly dividend from USD 1.24 to USD 0.82 per share. |
| 2015 March | GE signed an agreement with the Egyptian government to supply the country with turbines and other technology. |
| 2015 April | GE Capital said it would sell assets valued at USD 200 billion. |
| 2015 October | GE Transportation signed a USD 2.6 billion deal to supply 1000 locomotives to India. |
| 2015 November | A USD 9.5 billion purchase of French transportation company Alstom's power business was made. |
| 2016 | Allegations GE misled investors about the underlying profitability of its long-term health care and power units were made. |
| 2016 January | GE agreed to sell its appliance business to Qingdao Haier for USD 5.4 billion. |
| 2016 August | GE acquires ShipExpress. |
| 2016 November | GE Digital acquires ServiceMax. |
| 2017 January | GE announced it would cut 12,000 jobs. |
| 2017 November | GE unveiled a broad restructuring and halved its quarterly dividend. |
| 2017 November/December | GE laid off affected employees. |



**Table 3.** *Cont.*

| Date | Events |
|---|---|
| 2018 January | Flannery announced a previously unforeseen USD 6.2 billion insurance charge. |
| 2018 June | Removed from Dow Jones Industrial Average. |
| 2018 October/November | H. Lawrence Culp to replace John Flannery as chair and CEO of the company. Culp moves aggressively to reduce GE's debt and divest unwanted stakes and subsidiaries. |
| 2020 March | Announcement that the Aviation unit began laying off 10 percent of its U.S. workforce. |
| 2020 April | The coronavirus pandemic dealt a USD 1 billion blow to cash flow in its industrial business. |
| 2020 December | GE agreed to pay a USD 200 million penalty to settle charges for disclosure failures in its power and insurance businesses. |
| 2021 March | GE announced a deal merging its GE Capital Aviation Services (GECAS) aircraft leasing unit with AerCap Holdings. |
| 2021 November | GE unveiled a plan to split into three independent public companies. |
| 2022 April | GE warned fiscal 2022 annual earnings were "trending toward the low end". |

### 3.7. Approach Taken for Outlier Compensation

The second stage of the model is required to make adjustments based on the knowledge of outliers from the first stage. The adjustments made allow the model to produce a compensated prediction for the next few days. The data correlation analyses show it is possible to predict up to two days in advance, and any further prediction has a low correlation to the input data. The model developed in this paper will therefore predict a single day in advance from the two input variables, the day closing stock price and the loss function for the day. A higher-order neural network is used with two input variables, two hidden layers and a single output. The first hidden layer contains four neurons, and the second hidden layer contains two; both are fully connected layers [17]. Figure 5 is a graphical demonstration of this network.

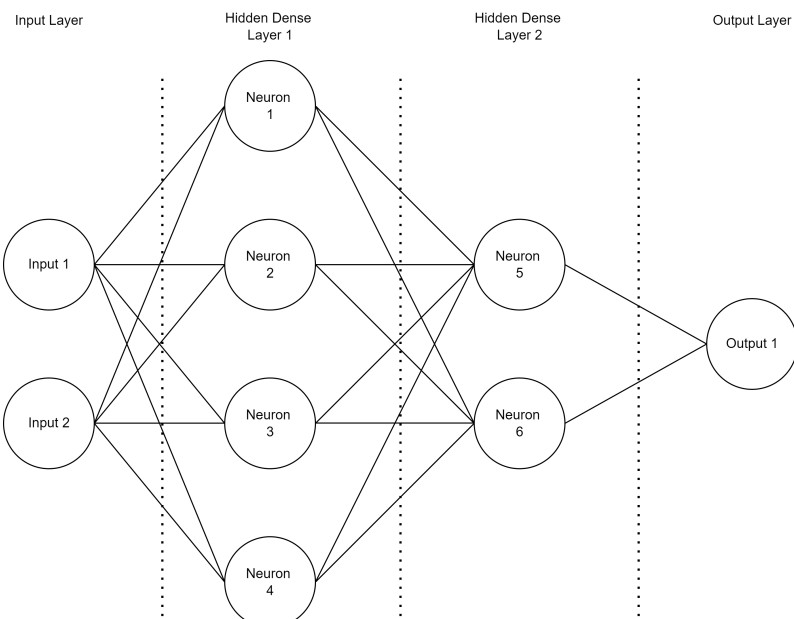

**Figure 5.** A high-level representation of the higher-order neural applied network.

From Tables A1–A3, the individual neuron functions of the feed-forward network are determined using Equation (7):

$$\text{Neuron} = f\left(\sum(\text{Weights} * \text{Inputs}) + \text{Bias}\right) \tag{7}$$

From Equation (7), each neuron could be seen to take in the inputs with an associated weighting. This is then summed with a common bias for the whole layer and finally, put through an activation function (in this case, ReLU—Rectified Linear Unit) to determine the value of that neuron. Equation (8) shows the individual neuron operations.

$$
\begin{aligned}
Y_1 &= \text{Relu}(W_{11}X_1 + W_{21}X_2) + B_1) \\
Y_2 &= \text{Relu}(W_{12}X_1 + W_{22}X_2) + B_2) \\
Y_3 &= \text{Relu}(W_{13}X_1 + W_{23}X_2) + B_3) \\
Y_4 &= \text{Relu}(W_{14}X_1 + W_{24}X_2) + B_4) \\
Y_5 &= \text{Relu}(W_{15}Y_1 + W_{25}Y_2 + W_{35}Y_3 + W_{45}Y_4) + B_5) \\
Y_6 &= \text{Relu}(W_{16}Y_1 + W_{26}Y_2 + W_{36}Y_3 + W_{46}Y_4) + B_6)
\end{aligned}
\tag{8}
$$

Given the values of each neuron, we may then determine the output of this neural network from Equation (9).

$$
Y_7 = \text{Relu}(W_{51}Y_5 + W_{61}Y_6) + B_7)
\tag{9}
$$

The corresponding algorithm for this stage is shown in Algorithm 2.

---

**Algorithm 2** A higher order neural network

---

**Require:** The closing stock price as of 1985; the probability of an outlier with magnitude and direction

  1: Initialization of required libraries;

  2: Load stock data;
  3: Load outlier data;
  4: Removal of invalid data:
  5: Scale data using the Keras Standard Scalar;
  6: Split data into training and test sets;

  7: Create a HONN model;
  8: Use the training data to train the HONN;
  9: Evaluate the model;

10: **while** Evaluation result improves **do**
11:     Adjust the hyper-parameters of the model;
12: **end while**

13: Remove normalisation of data;
14: Produce a final compensated prediction of the closing stock price;

---

The algorithm will be implemented on top of the TensorFlow library using Keras. The model was executed using Google Colab with hardware acceleration using a Tesla K80 GPU. The processing time required (*s*) to process the data was 0.767 *s*; the time required to train the model was 41.243 *s*, and the time required to execute the model was 1.108 *s*. Our implementation method was not intended to perform a computational complexity analysis of the algorithm but to allow for the evaluation of the concept. The model code for the higher-order neural network is shown in Listing 2.

Listing 2: HONN model

```
model2 = Sequential()
# HONN input layer
model2.add(Dense(4, input_dim = 2, activation = 'relu'))
```

```
#HONN Hidden layers
model2.add(Dense(2, activation = 'relu'))

#HONN output layer
model2.add(Dense(1, activation = 'relu'))

#Optimizer and loss function
optim = tf.keras.optimizers.SGD(learning_rate = 20e-4)
model2.compile(optimizer = optim, loss = 'mse')

model2.summary()
```

The resultant model parameters of the HONN are shown in Table 4.

**Table 4.** The higher-order neural network model summary.

| Layer (Type) | Output Shape | Parameters |
|---|---|---|
| Dense 1 | (None, 4) | 12 |
| Dense 2 | (None, 2) | 10 |
| Dense 3 | (None, 1) | 3 |

The model parameters shown in Table 4 show the type of layer, the shape of the layer and the number of editable parameters in each layer. The parameters were adjusted during the training phase to match the data features.

*3.8. Environment*

The method proposed was applied in Google Colaboratory, a product offered by Google Research that takes the form of a hosted Jupyter notebook for machine learning, data analysis, etc. This platform requires little to no setup to use while providing access to computing resources.

**4. Results**

The occurrence of outlier events can be seen to have a significant effect on the trajectory of financial time series data, which has increased the difficulty of timely prediction. The approach used must be able to detect outlier events that have occurred in financial time series data and compensate for it in its prediction.

*4.1. LSTM Auto-Encoder Experiments and Results*

4.1.1. Training of the LSTM Auto-Encoder

To train the LSTM auto-encoder model, a limit of 200 epochs was defined with early stopping criteria based on the validation loss. An epoch is an iteration in the training process; each epoch is a single pass over the batched training data [32]. The validation loss was monitored for a minimum of over three epochs. If the validation loss did not decrease during this period, the training was halted. The minimum validation loss achieved after training the device was 0.0729. The convergence of the training process is shown in Figure A1, where the training process was run over a maximum of 200 epochs. After training the LSTM auto-encoder, the model was then validated against the test data set, unseen from the initial training process, and the validation loss was found to be 0.043.

4.1.2. Output from the LSTM Auto-Encoder

The LSTM auto-encoder produced a reconstructed sequence of the original input. The errors in this recreated sequence were used for the determination of an anomalous event. Figure 6 shows the ground truth sequence against the reconstructed sequence.

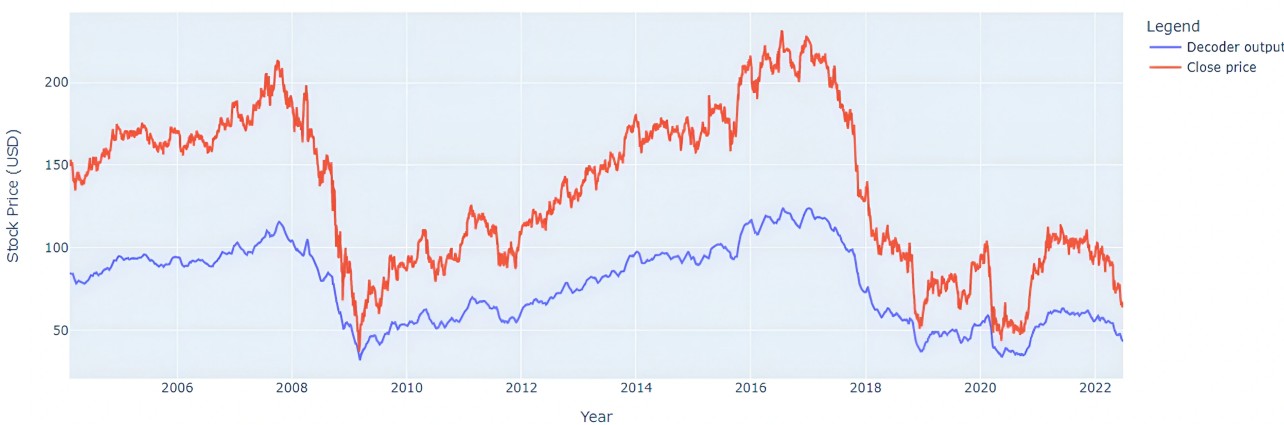

**Figure 6.** Output sequence of the LSTM auto-encoder.

To determine anomalies, a threshold was found from the training loss distribution shown in Figure 7.

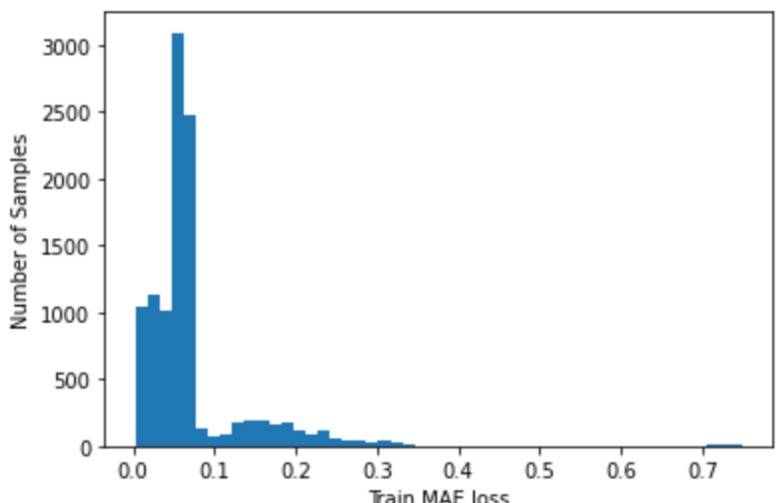

**Figure 7.** The training loss distribution of the LSTM auto-encoder.

Given the distribution of the data, its statistical mean, median, mode standard deviation and variance can be determined [33]. The threshold is set as the mean of the distribution summed with three standard deviations to find values that do not conform with the normal distribution (i.e. not matching 99.7 percent of the distribution) [33]. The defined threshold can now be applied to the test loss distribution to determine anomalies as shown in Figure 8.

The anomalies that have been detected by comparing the test loss against the threshold found are shown in Figure A3. The figure shows the closing price of General Electric plotted with each outlier superimposed onto this plot with a red dot. Typical stock outlier detection methods will have a greater number of detected events because of their rigid rule set.

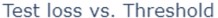

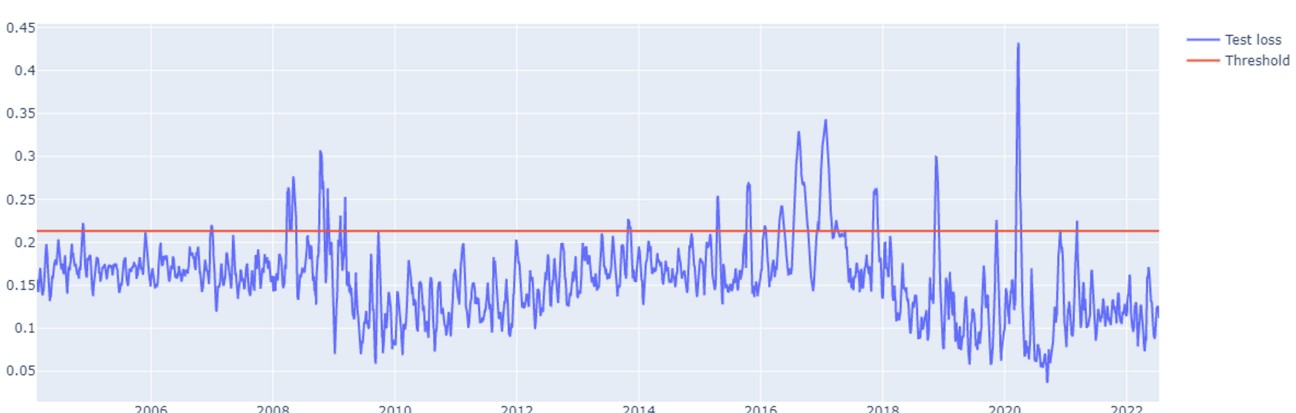

**Figure 8.** The test loss and training threshold from the LSTM auto-encoder.

*4.2. HONN*

4.2.1. Training of the HONN

To train the higher-order neural network, a limit of 150 epochs was defined without early stopping criteria enabled; the converging results of the training are shown in Figure A2. Upon completing the training, the minimum validation loss of the training data set was found to be 0.0074. The trained model was then validated against unseen data from the initial training split resulting in a validation loss of 0.0021.

4.2.2. Output from the HONN

The HONN produces a compensated prediction from the stock closing price and the loss function of the LSTM auto-encoder, which is the basis for compensating an outlier to improve forecast accuracy. The forecast made is plotted against the ground truth price in Figure 9 with the anomalous areas marked.

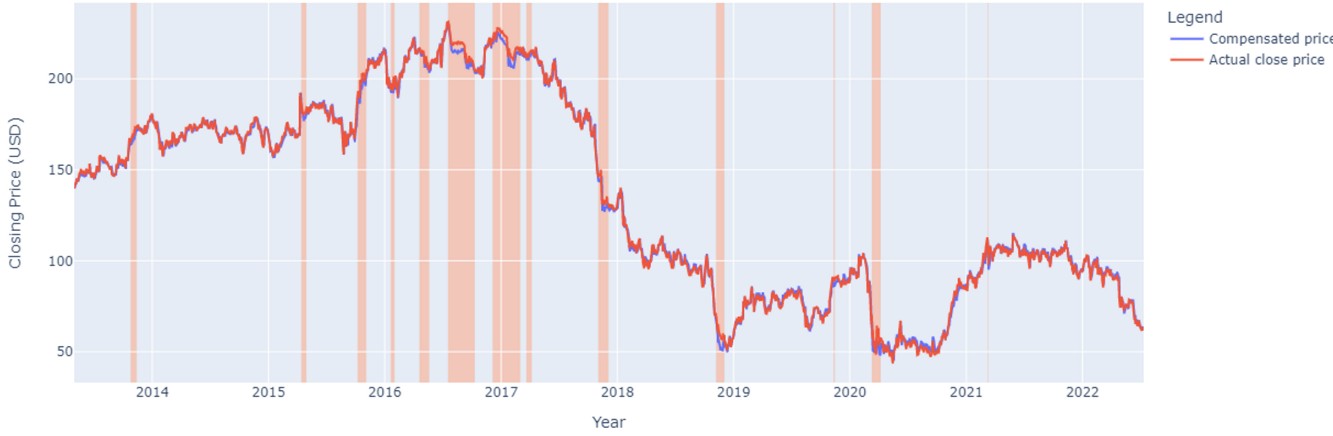

**Figure 9.** Compensated prediction of the HONN compared to the ground truth stock price with marked anomalous areas.

4.2.3. Comparison to State-of-the-Art Methods

The approach was validated using two recurrent deep learning architectures shown to be adept at time series forecasting. The first validation model used was a deep LSTM model,

which consisted of two LSTM layers and two dense layers. The model architecture was adapted from a 2017 paper on LSTM [22]. The summary of the model is shown in Table 5. The model was optimised using an adaptive moment estimation (ADAM) optimizer and trained for 120 epochs. Upon evaluating the model against unseen data, the model was found to have a loss of 0.0194.

**Table 5.** The validation LSTM model summary.

| Layer (Type) | Output Shape | Parameters |
|---|---|---|
| LSTM 1 | (None, 30, 128) | 66,560 |
| Dropout 1 | (None, 30, 128) | 0 |
| LSTM 2 | (None, 64) | 49,408 |
| Dropout 2 | (None, 64) | 0 |
| Dense 1 | (None, 16) | 1040 |
| Dropout 3 | (None, 16) | 0 |
| Dense 2 | (None, 1) | 17 |

The second validation model used a deep GRU architecture consisting of four GRU Layers and a single dense layer. The model architecture was adapted from a 2020 paper on GRU [24]. The model was optimised using stochastic gradient descent and trained for 100 epochs. Upon evaluating the model against unseen data, the model was found to have a loss of 0.0105. The model summary is shown in Table 6.

**Table 6.** The validation GRU model summary.

| Layer (Type) | Output Shape | Parameters |
|---|---|---|
| GRU 1 | (None, 30, 50) | 7950 |
| Dropout 1 | (None, 30, 50) | 0 |
| GRU 2 | (None, 30, 50) | 15,300 |
| Dropout 2 | (None, 30, 50) | 0 |
| GRU 3 | (None, 30, 50) | 15,300 |
| Dropout 3 | (None, 30, 50) | 0 |
| GRU 4 | (None, 50) | 15,300 |
| Dropout 4 | (None, 50) | 0 |
| Dense | (None, 1) | 51 |

The outputs from the validation models are shown in Figure 10 against the ground truth stock closing price, with the anomalous areas marked. The validation models can be seen to handle the data features well, particularly from 2018 onwards; this can be attributed in part to fewer anomalous regions. The LSTM model can be seen to better handle the core pattern void of outliers, while the GRU model is more resilient against anomalies. Despite the resilience of the GRU model, it still cannot effectively model the effect that an anomalous event will have on a stock price.

**Figure 10.** Output prediction of the proposed method compared to the validation models and ground truth stock price.

The output from the proposed method is included in Figure 9 against the output of the validation models and the ground truth stock price, with anomalous areas marked. The proposed model can be seen to be far more reactive to an anomalous event in both amplitude and direction, which are key to the mitigation of associated risks. To quantify this phenomenon, the following validation equations are used, based on determining the accuracy of stock prediction models [34] and time series accuracy measures [35]: mean absolute error, as shown in Equation (10); mean square error, as shown in Equation (11); and mean absolute percentage error, as shown in Equation (12).

$$\text{MAE} = \frac{1}{n} \sum_{i=1}^{n} |x_i - y_i| \tag{10}$$

We consider the mean absolute error (MAE), which provides us with an absolute measure of the errors between forecasts and the observed value. This parameter excludes the direction of the error from the calculation.

$$\text{MSE} = \frac{1}{n} \sum_{i=1}^{n} (x_i - y_i)^2 \tag{11}$$

The mean square error (MSE) provides us with a measure of the severity of errors in the forecasts made. This calculation excludes the direction of the error but amplifies its magnitude.

$$\text{MAPE} = \frac{1}{n} \sum_{i=1}^{n} \left| \frac{x_i - y_i}{x_i} \right| \tag{12}$$

The mean absolute error (MAPE) is the percentage error of a forecast relative to the actual value; this equation gives equal weight to all instances. A lower percentage error is preferable in this scenario.

$$\sigma = \sqrt{\frac{1}{N-1} \sum_{i=1}^{N} (x_i - \overline{x})^2} \tag{13}$$

We determine the standard deviation of the error to verify the spread. A smaller spread indicates that the forecasts made are close to the ground truth values.

The key validation results have been tabulated in Table 7. This shows the propensity of the proposed model to adjust to outliers. The proposed method has a lower error rate and error distribution than the validation models.

**Table 7.** Results of methods against unseen data.

| Predictor | Mean Absolute Error | Mean Square Error | Mean Absolute Percentage Error | Error Standard Deviation |
|---|---|---|---|---|
| LSTM | 0.712 | 0.770 | 3.993% | 0.148 |
| GRU | 0.717 | 0.779 | 4.331% | 0.144 |
| The proposed method | **0.034** | **0.002** | **0.030%** | **0.030** |

In Table 8, the results have been collected from the 2019 International Conference on Intelligent Computing [36]. Only results with equivalent error indicators have been used. From the results, a deep learning method used on the Chinese securities index 10 had a MAPE of 0.0002%. Our proposed method comes in second in this regard; however, the root-mean-square error (RMSE) of the aforementioned deep learning method is 1.0437. Our proposed method has an RMSE of 0.0447. This indicates that our method is better grouped around the true value, showing the affinity of the method to compensate for outliers.

**Table 8.** Results of the proposed method against published results of prediction methods.

| Data Set | Method | Mean Absolute Percentage Error |
|---|---|---|
| Chinese securities index 10 | Deep Learning | **0.0002%** |
| S&P Global | FLIT2FNS | 0.3200% |
| S&P 500 | LS-Random Forest | 0.5745% |
| S&P 500 | LSTM | 0.7240% |
| S&P 500 | Particle Swarm Optimization | 0.6558% |
| BSE India | SVM-KNN | 0.1123% |
| BSE India | FLIT2FNS | 0.6100% |
| BSE India | CEFLANN | 1.8000% |
| BSE India | SVR-ANN | 2.6600% |
| Goldman Sachs | Regression | 1.4726% |
| NYSE | CNN | 5.3100% |
| NYSE | Proposed method | 0.0300% |

## 5. Discussion

### 5.1. Summary of Findings

The occurrence of outlier events has had a net effect on the trajectory of financial time series data. This effect is confirmed by being able to compensate for outliers to make a more accurate prediction. A deep learning method to detect outliers is a feasible approach; however, the model must be fine-tuned to the stock we intend to model and then validated against labelled events in a semi-supervised learning approach. The traditional time series prediction methods typically use some form of gated recurrent neural networks. State-of-the-art versions of this type of network have been used to validate the compensated approach made in this paper to show any improvements in accuracy.

The selection of technical variables for time series prediction should be accomplished via time-dependent correlation and feature engineering. This will allow for the selection of variables that have a strong direct and lagged correlation. The detection of outliers was done through an auto-encoder that has been adapted to time series data with an LSTM. Auto-encoders were found to be able to find the core pattern in the data through dimensionality reduction. The verification of the results against labelled data is crucial to minimise false detections.

The use of multiple models was essential to the successful detection and compensation of outlier events. The result of the compensation model is completely dependent on the

quality of the outlier detection model. By evaluating the training and validation results, the models were shown to be able to correctly model the core patterns in the stock data and detect outlier events correctly. For the successful use of multivariate data for time series prediction, the data should share some correlation between the input variables and the output. Input variables that do not share any correlation between the two may be used but do not offer as high a resultant model accuracy. If a single day in advance is being predicted, a higher-order neural network may be applied as opposed to a recurrent neural network and is more accurate.

### 5.2. Implication of Findings

It is possible to detect outlier events using deep learning and then improve the prediction accuracy of a model using the events detected. This may be expanded to traditional outlier detection methods in the financial sector. An improvement in the detection of market manipulation and an improvement in predictions will allow for a more stable stock market and by extension, the economy.

### 6. Conclusions

This study proposed the use of a deep learning method to detect and compensate for anomalies in financial time series forecasting. The feasibility of our proposed method is supported by the results. Existing methods for predicting financial time series were used to validate our method. Our approach achieved a mean absolute percentage error of 0.03% and an error standard deviation of 0.03. The error indicators used show the predictions generated by our approach are consistently close to the true value. The results of our approach are significantly lower than those achieved by the validation methods, and our method additionally performs well against published methods on different data sets. An advantage of our methodology is that it only requires the use of labelled anomalous periods to fine-tune the threshold, allowing it to be applied generally to any stock. Additional fine-tuning can be made to detect anomalies that are not obvious but will require a completely unsupervised approach. The compensation approach used has shown that a compensated model is better suited to handling anomalous periods and adjusting its prediction to closely match the true value. This is as compared to a model without any compensation. The results support our hypothesis that it should be possible to detect outlying events and therefore improve forecasting using these detections. Previous studies only attempted to detect outliers using a similar LSTM approach. These studies do not explore the possibility of improving forecasts based on these detections. Future research should be considered to extend the forecasting window beyond two days (two periods). This may entail the evaluation and optimisation of other deep learning methodologies.

### Delimitation of Methodology

Our methodology sought to determine if it is indeed possible to compensate for outliers in a prediction machine learning model. Our approach does, however, suffer from some shortcomings. Our current architecture will only allow the model to predict up to two days in advance, as there is a steep drop off in accuracy as the prediction window is extended. The extension of our approach could be applied to other stocks; however, the architecture of the compensation model will need to be changed on a per-stock basis. This limitation prevents the creation of a generally trained model that could be used as a basis for other stocks. A lack of availability of quality labelled events will hamper the performance of the method and may result in the model either over-compensating or under-compensating for outliers in its prediction.

**Author Contributions:** Writing—original draft preparation, V.N.; writing—review and editing, V.N. and S.D.; supervision, S.D. All authors have read and agreed to the published version of the manuscript.

**Funding:** This research is funded by the South African National Research Foundation Incentive Grants (No. 145975).

**Institutional Review Board Statement:** The study was conducted in accordance with the submission for ethics approval, approved by the Institutional Review Board (or Ethics Committee) of the Faculty of Engineering and the Built Environment Research Ethics Committee, and assigned the approval number FCRE2021/11/008-EBE.

**Data Availability Statement:** The source code is available from our Github repository https://github.com/vashalenmn/Eng_Project_2020.git, accessed on 21 September 2022. All source data for General Electric are available from https://finance.yahoo.com/quote/GE/ accessed on 21 September 2022 (closing price); event classification is available from www.ge.com/news accessed on 16 June 2022 and https://www.bloomberg.com/graphics/2019-general-electric-rise-and-downfall/ accessed on 16 June 2022.

**Conflicts of Interest:** The authors declare no conflict of interest. The funders had no role in the design of the study; in the collection, analyses, or interpretation of data; in the writing of the manuscript, or in the decision to publish the results.

## Abbreviations

The following abbreviations are used in this manuscript:

| | |
|---|---|
| ADAM | adaptive moment estimation |
| AI | Artificial Intelligence |
| ANN | Artificial neural network |
| API | Application Programming Interface |
| ARIMA | Auto-Regressive Integrated Moving Average |
| EMH | Efficient Market Hypothesis |
| GPU | Graphics Processing Unit |
| GRU | Gated recurrent unit |
| HONN | Higher order neural network |
| LSTM | Long-short term memory |
| NN | Neural network |
| NYSE | New York Stock Exchange |
| ReLU | Rectified Linear Unit |
| SGD | Stochastic gradient descent |

## Appendix A

*Appendix A.1*

**Table A1.** Corresponding weights and biases between the input layer and hidden layer 1.

| | Input 1 | Input 2 | Biases |
|---|---|---|---|
| Neuron 1 | $W_{11}$ | $W_{21}$ | $B_1$ |
| Neuron 2 | $W_{12}$ | $W_{22}$ | $B_2$ |
| Neuron 3 | $W_{13}$ | $W_{23}$ | $B_3$ |
| Neuron 4 | $W_{14}$ | $W_{24}$ | $B_4$ |

*Appendix A.2*

**Table A2.** Corresponding weights and biases between hidden layer 1 and hidden layer 2.

| | Neuron 1 | Neuron 2 | Neuron 3 | Neuron 4 | Biases |
|---|---|---|---|---|---|
| Neuron 5 | $W_{15}$ | $W_{25}$ | $W_{35}$ | $W_{45}$ | $B_5$ |
| Neuron 6 | $W_{16}$ | $W_{26}$ | $W_{36}$ | $W_{46}$ | $B_6$ |

*Appendix A.3*

**Table A3.** Corresponding weights and biases between the output layer and hidden layer 2.

|          | **Neuron 5** | **Neuron 6** | **Biases** |
| -------- | ------------ | ------------ | ---------- |
| Output 1 | $W_{51}$     | $W_{61}$     | $B_7$      |

*Appendix A.4*

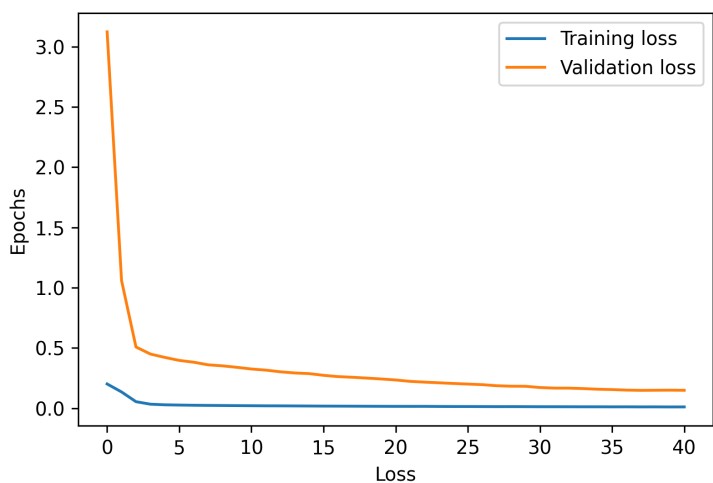

**Figure A1.** Training and validation losses during the training of LSTM auto-encoder.

*Appendix A.5*

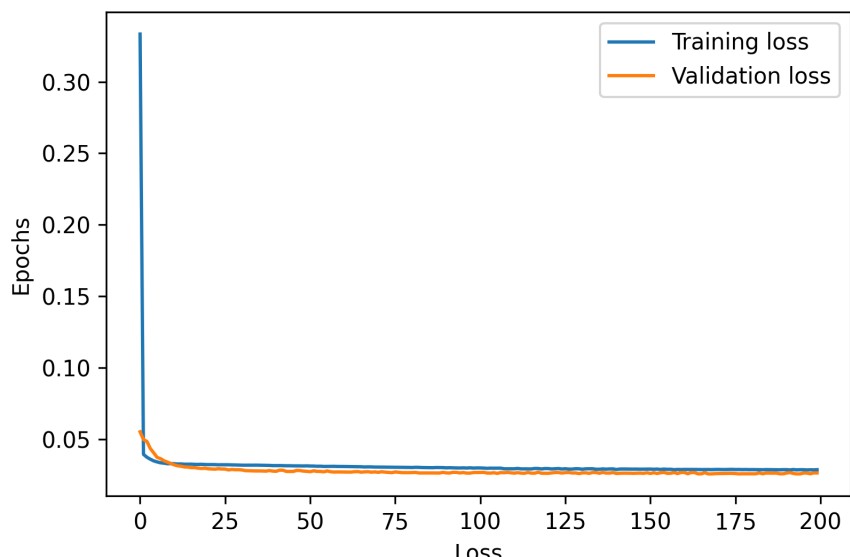

**Figure A2.** Training and validation losses during the training of HONN.

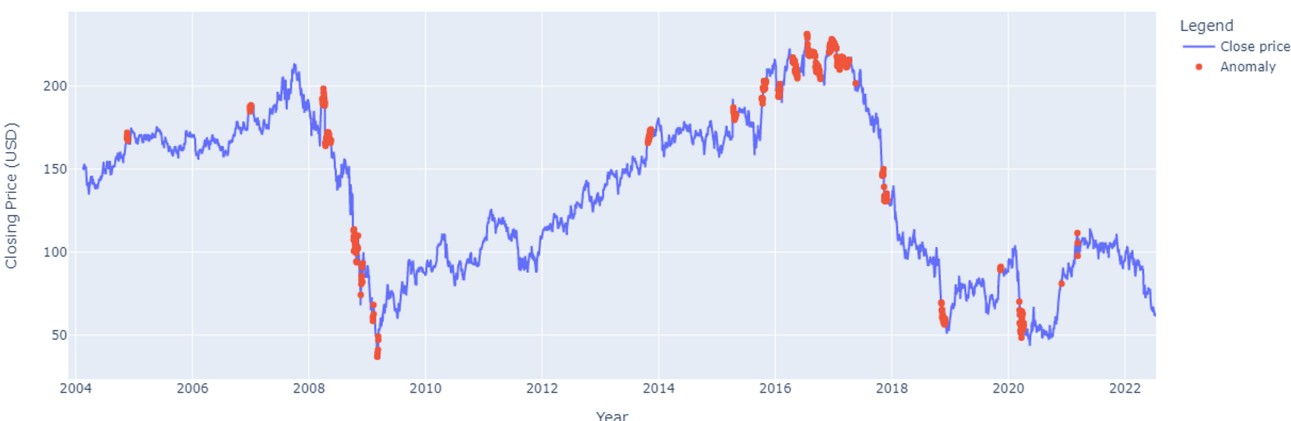

**Figure A3.** The ground truth stock price with the detected anomalies marked.

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
