# Peer review of "A Deep Learning Method for the Detection and Compensation of Outlier Events in Stock Data"

_electronics, doi:10.3390/electronics11213465_

Round 1

Reviewer 1 Report (Previous Reviewer 1)

A DEEP-LEARNING METHOD FOR THE DETECTION AND

COMPENSATION OF OUTLIER EVENTS IN STOCK DATA

I would like to suggest few modifications to the existing manuscripts as per following.

1. In the introduction part contributions should be highlighted in points.

2. Related work section should have tabular comparisons in all subsections such as 2.1 to 2.12 in place of general theoretical discussions.

3. Re-design with more details, all figures numbering from 1 to 4.

4. Add a flowchart of the complete mechanism with appropriate notations etc.

5. Make separate section and analyse the complexity of the proposed framework in details.

6. Result section needs revision and specifically highlights the important achievements.

7. Finally conclusion needs revision.

8. Add future research directions and possible drawbacks of the present approach.

Author Response

Thank you for your suggestions, your honest opinions have allowed for me to improve the manuscript immeasurably for the benefit of the machine learning industry and academic institutions as a whole.

Kind regards,

V Naidoo

Reviewer 2 Report (Previous Reviewer 3)

Dear Author,

Thank you for your contribution.

You can see the original message to the Editorial Board below:

<<

Honored Editorial Board,

As the title, abstract and content suggest this paper proposes a deep-learning methodology for detecting and compensating for outlier events in stock data.

After reading the entire manuscript, I found that there are still some issues needing further attention (to consider and solve).

I will start with the format ones. Then I will continue with those related to the content/substance of the manuscript.

For the first class (format issues), I mention below:

  • English language and style issues - Grammarly (https://app.grammarly.com) on default settings (American English, Set Goals: Audience=Knowledgeable, Formality=Neutral, Domain=General) detected only for the text block resulting from the concatenation of Title+Abstract+Keywords+Conclusion (By the way, the plural is needed here because there are many conclusions in the corresponding paragraph):

(a) 2 correctness issues / critical alerts

but

(b) 25 more advanced ones, namely Word choice (9), Punctuation in compound/complex sentences (5), Unclear sentences (4), Passive voice misuse (3), Wordy sentences (3), and more (1).

The resulting Grammarly's total score was 85 (a good one, in my opinion, but there is room for more) out of 100 (max) for this four-component sample above.

Still, since the author does not appear to be a native English speaker, I suggest a moderate revision of the English language and style for this article using Grammarly or another specialized tool;

  • Except for minor differences, the paper mostly follows the specific structure of the journal, namely:
    Author Information, Abstract, Keywords, Introduction, Materials & Methods, Results, Discussion, Conclusions, etc., as indicated at: https://www.mdpi.com/journal/electronics/instructions;

  • All reference to the equations/formulas must be explicitly and precisely formulated in the main text (e.g. “see eq.X”);

  • The figures with charts representing data (starting with Figure 6) need more resolution or extra settings for higher resolution when exporting to the .pdf format. First, the author must ensure that all figures have the required resolution (minimum 1000 pixels width/height, or a resolution of 300 dpi or higher according to the Journal’s instructions at the link above);

  • Overall, there are so many figures (13) and tables (9) included in the main content. A special section (Appendix - at the end of the manuscript) must be included and those figures and tables considered not essential for understanding the main ideas must be moved there;

  • The authors must include the DOI codes for most journal references.

For the second class of issues (content-related ones), I mention below the following:

  • I think more contributions in journal papers must be cited in this research both in the Introduction and especially in the section dedicated to the interpretation of the results. I also think that just 32 references (from which 3 are proceedings and one - symposium) are not enough according to the requirements of this journal;

  • Additional references to scientific papers are also needed when introducing the concept of “epoch”;

  • If those two occurrences of the “Stock Dataset” on the left of figure 5 refer to different datasets/subsets, the author must use 1 and 2 at the end of these names or indicate precisely if it is about the same dataset;

  • Most of the existing figures need extra interpretation in relation to similar/divergent results (existing and published in papers with related content);

  • The limitations of this methodology must be specified in a dedicated section at the end of the manuscript;

  • After isolating Limitations from Conclusions, the latter must be extended to include more of the author’s contributions;

  • Sections related to funding, data availability, and acknowledgments are available in this manuscript, which is a great thing;

  • Additional references to scientific papers must be included when commenting on results in order to sustain the arguments regarding the intervals for the values and the allowed thresholds if applicable (e.g. after Table 9);

  • Algorithms 1 (page 9) and 2 (page 14) must be precisely identified in the authors’ own GitHub repository (replication of results reason). If not existing, the authors must create one for the entire project corresponding to this manuscript.

Thank you for the opportunity to read and check this contribution!

>>

Author Response

Thank you for your suggestions, your honest opinions have allowed for me to improve the manuscript immeasurably for the benefit of the machine learning industry and academic institutions as a whole.

Kind regards,

V Naidoo

Reviewer 3 Report (Previous Reviewer 2)

This paper proposes a deep learning method that uses two sequential stages. The first stage is an outlier detection model using a long short-term memory (LSTM) network auto-encoder that can determine if an outlier event has occurred and then create an associated value of this occurrence for the next stage. The second stage of the proposed method uses a Higher Order Neural Network (HONN) model to make a prediction based on the output of the first stage and the stock time series data.

(1) In the abstract, the author should highlight the specific problems to be solved in this study at the beginning, and then lead to the solutions. At present, the description is not clear.

(2) In the introduction, it is necessary to add a research background introduction and a detailed explanation of the research motivation, so as to attract more potential audiences.

(3) The method/approach in the context of the proposed work should be written in detail.

(4) The computation complexity of the proposed method should be clearly described.

(5) In Section 2, the basic methods and theories are too lengthy. Delete some methods and theories that are not the core, such as Machine learning algorithms, and so on.

(6) What are the difference and  relation between the outlier detection Anomaly Detection at Lines 253-267?

(7) The literature review is poor in this paper. You must review all significant similar works that have been done. For example, https://doi.org/10.3390/agriculture12060793; https://doi.org/10.1109/JSTARS.2021.3059451 and https://doi.org/10.1016/j.ins.2022.08.115 and so on.

(8) There are some grammatical errors seen in the paper. Check carefully for a few clerical errors and formatting issues. 

Author Response

Thank you for your suggestions, your honest opinions have allowed for me to improve the manuscript immeasurably for the benefit of the machine learning industry and academic institutions as a whole.

Kind regards,

V Naidoo

Round 2

Reviewer 1 Report (Previous Reviewer 1)

The manuscript in its current state may be considered for publication subject to complete proofreading and removal of grammatical mistakes etc..

Reviewer 3 Report (Previous Reviewer 2)

The paper can be accepted now. 

This manuscript is a resubmission of an earlier submission. The following is a list of the peer review reports and author responses from that submission.

Round 1

Reviewer 1 Report

A DEEP-LEARNING METHOD FOR THE DETECTION OF OUTLIER EVENTS FROM STOCK DATA- 1863924

I would like to highlight a few facts.

1. Starting from the introduction part, this part is poorly written and where is the list of significant contributions? Also, information about the organization of the complete text is also missing.

2. Unable to find the related work section?

3. Unable to find the flow chart of the methodology?

4. No mathematical modelling with equations.

5. Complexity analysis is the main part of these scenarios, unable to find in the text.

6. System diagram is not clear, only textual information is written. Figure 1 is also not clear.

7. Unable to find any sufficient reasons for selecting this particular approach?

8. Reference section is poorly written with only 12 citations.

Overall, the lack of novelty is the main reason for the rejection of this research work.

Reviewer 2 Report

 The results look encouraging and motivating. But there are still some contents, which need be revised in order to meet the requirements of publish. A number of concerns listed as follows:

(1)   The abstract does not provide significant information and it should be revised to highlight the significant methodological contributions and conclusions.

(2)   In the Section, the main contributions of this paper should be further summarized and clearly demonstrated.

(3)   The author should provide the organization of the article in the introduction so that the readers should understand the workflow easily.

(4)   Figures quality need to be enhanced to better understanding.

(5)   The methodology is not clear and it can be further improved it is better to add a flow chart of methodology.

(6)   The results can be generalized in present form some results are redundant. Compared with the existing methods, the innovation of the proposed method needs more detailed description.

(7)   Please provide a flow of the proposed semi-supervised transfer learning methodology.

(8)   All abbreviations need to be written in full for the first time, such as Line 53, LSTM, Line 60, NYSE,  and so on.

(9)   Result and discussion should be rewritten to summarize the significance of the work.

(10) In order to highlight the introduction, some latest references should be added to the paper for improving the reviews part. For example, 10.1016/j.isatra.2021.07.017 ;10.1109/JSTARS.2021.3059451 ; 10.3390/agriculture12060793 ï¼› 10.1007/s10489-022-03719-6 and so on.

(11) What are the advantages and disadvantages of this study compared to the existing studies in this area in the Section of Conclusion?

Reviewer 3 Report

Dear Authors,

After reading your paper, I came to the following conclusions regarding still existing issues you should deal with.

  1. I will start with the ones related to language and style, namely:

  • The Grammarly online app. (on default settings)

detected for a text sample meaning the concatenation of Title+Abstract+Keywords+Conclusions

the following issues:
(a) 12 critical alerts/correctness issues and
(b) 34 more advanced ones, namely:

->Word choice (11 issues),

->Wordy sentences (7 issues),

->Unclear sentences (5 issues),

->Passive voice misuse (5 issues),
->Hard-to-read text (2 issues),

->and more (4 issues).
For this sample above, Grammarly returned a score of 67 out of 100 (maximum). In addition, you do not appear to be native English-speaking authors. Therefore, I have chosen (in the review form) the option corresponding to Extensive Editing of English Language and Style Required for the entire article using a specialized tool (e.g., Grammarly or MDPI’s online service).

  1. I will continue with other format-related issues:

  • The paper should precisely follow the specific structure of the journal, namely:
    Author Information, Abstract, Keywords, Introduction, Materials & Methods, Results, Discussion, Conclusions, etc., as indicated at: https://www.mdpi.com/journal/electronics/instructions. For instance, the Discussion is missing. The Materials & Methods was renamed “Method for outlier detection and compensation”. 

  • In the same format terms, you must ensure that all figures have the required resolution (minimum 1000 pixels width/height, or a resolution of 300 dpi or higher according to the same Journal’s instructions at the link above. I think there is room for more resolution (both dpi and pixels on both axes) in Figures 1, 3, 5

  • Also, on the matter of format, I think figures 2, 4, 6, 12, 15, and 16 should be transformed into tables;

  • In the same matter, I think you must avoid ending some sections/subsections with figures/tables  (e.g. Figure 4 just before subsection 2.5. - Environment; Table 5 just before Conclusions). I think they can move it between text blocks or add some explanatory text after. The same for Figures 5, 11, and 14;

  • The equation/formula below line 189 must be properly numbered and precisely referenced (e.g., - see eq.N..) in the main text (the same Journal’s instructions at the link above);

  1. I will continue with content-related observations:

  • I appreciate the inclusion of the algorithms and the data availability (support for replicability in science), the list of abbreviations, and the existence of the ethics approval;

  • I do not appreciate such a tiny list of references meaning that a lot of previous related contributions in journal papers are most probably not cited. And this should be solved (is a must) in this research in both the Introduction and/or Literature Review section and the one dedicated to the interpretation of the results (Discussion - which is missing, by the way) together with comparisons to already existing results in this specific field;

  • You are required to properly cite the reference in line 287 (both format and content issue). The same for line 295;

  • I think you meant Mean Error and Standard Error in the header of Table 5. You are asked to perform the corresponding replacements (No “of”);

  • What about some other time-series accuracy measures (not included in Table 5)? You are asked to consider useful scientific resources at: https://scholar.google.com/scholar?hl=ro&as_sdt=0%2C5&q=%22time+series+accuracy+measures%22&btnG= and improve the content of Table 5;

  • In the same content terms, you are required to better underline their own contributions in the Conclusions section;

  • In the same content matter, ARIMA is missing from the list of abbreviations. You are required to solve this particular issue and check the rest of the manuscript for other similar ones;

  • You are required to explain in simple words (provide more details in the manuscript) what exactly they mean by epochs (line 289);

  • The lack of Discussion and the poor list of references (only 12 from which 4 are online materials) give the feeling of unfinished research (to be honest, in at least an early stage).

 So my decision is indulgently a "major revision" (read as a lot of major revisions).

Thank you for your contribution!

Sincerely,

D.H.